# Effect of Microcystin-LR, Nodularin, Anatoxin-a, β-N-Methylamino-L-Alanine and Domoic Acid on Antioxidant Properties of Glutathione

**DOI:** 10.3390/life12020227

**Published:** 2022-01-31

**Authors:** Michal Adamski, Ariel Kaminski

**Affiliations:** 1Department of Phycology, W. Szafer Institute of Botany, Polish Academy of Sciences, Lubicz 46, 31-512 Kraków, Poland; 2Laboratory of Metabolomics, Faculty of Biochemistry, Biophysics and Biotechnology, Jagiellonian University, Gronostajowa 7, 30-387 Kraków, Poland

**Keywords:** cyanobacteria, cyanotoxins, glutathione, antioxidant properties

## Abstract

Cyanobacteria produce a range of toxic secondary metabolites that affect many processes in human, animal and also plant cells. In recent years, some efforts have concentrated on deepening the understanding of their effect on living cells in the context of the disruption of antioxidant systems. Many results suggest that cyanotoxins interfere with glutathione (GSH) metabolism, which often leads to oxidative stress and, in many cases, cell death. Knowledge about the influence of cyanotoxins on enzymes involved in GSH synthesis or during its antioxidant action is relatively broad. However, to date, there is no information about the antioxidant properties of GSH after its direct interaction with cyanotoxins. In this paper, we investigated the effect of four cyanotoxins belonging to the groups of hepatotoxins (microcystin-LR and nodularin) or neurotoxins (anatoxin-a and β-N-methylamino-L-alanine) on the in vitro antioxidant properties of GSH. Moreover, the same study was performed for domoic acid (DA) produced by some diatoms. The obtained results showed that none of the studied compounds had an effect on GSH antioxidant potential. The results presented in this paper are, to the best of our knowledge, the first description of the kinetics of scavenging radicals by GSH reactions under the influence of these cyanotoxins and DA. This work provides new and valuable data that broadens the knowledge of the impact of cyanotoxins and DA on GSH metabolism and complements currently available information. Future studies should focus on the effects of the studied compounds on antioxidant systems in vivo.

## 1. Introduction

Cyanobacteria and diatoms are able to produce a large number of substances with high biological activity. Certain secondary metabolites synthesized by cyanobacteria, known as cyanotoxins, are some of the most dangerous compounds produced in nature. Based on their effect on vertebrates, these compounds can be divided into four groups: hepatotoxins, neurotoxins, cytotoxins and dermatotoxins [1,2,3,4,5].

The most well characterized group are hepatotoxins, represented by cyclopeptides microcystins (MCs), including microcystin-LR (MC-LR) (Figure 1A) and nodularin (NOD) (Figure 1B).

Conveyed to the liver via bile acid transporters, the harmful effects of MCs on hepatocytes develop rapidly [6,7]. Intracellularly, these compounds are strong inhibitors of protein phosphatases 1 and 2A and are potentially cancerogenic [8,9,10]. MCs also induce the excessive production of reactive oxygen species (ROS) that react with elements of the cytoskeleton and disrupt its function [11,12]. The consequences of its effects on the liver are hemorrhages that precede excessive blood loss and lead to organ failure [13,14]. They are similar in chemical structure to MCs (differing only in the number of amino acids) as well as having similar bioactivity to another cyanotoxin, NOD [15].

Among neurotoxins is the alkaloid anatoxin-a (ANTX-a) (Figure 1C) and non-protein amino acids: synthesized mainly by cyanobacteria β-N-methylamino-L-alanine (BMAA) (Figure 1D) and produced by some diatoms’ domoic acid (DA) (Figure 1E). ANTX-a is a cholinergic agonist that binds to nicotinic acetylcholine receptors in nerves and at neuromuscular junctions resulting in death due to suffocation [16]. The action of BMMA is slower, and symptoms of its toxicity can potentially develop over many years [17,18]. An example of the influence of BMAA on the human nervous system is a disease named amyotrophic lateral sclerosis-parkinsonism-dementia (ALS-PDC) that affected the indigenous people of the island of Guam. BMAA most likely accumulated in cells after consuming products made from the seeds of cycads that have a symbiotic relationship with cyanobacteria from the genus *Nostoc* and contain the toxin. Moreover, an element of the diet of the Guam people was traditional bat soup, and the tissues of these animals were additional sources of BMAA (bats consumed cycads seeds). Symptoms of ALS-PDC are similar to those observed in Parkinson’s and Alzheimer’s diseases [19,20]. At the cellular level, the impact of BMAA on the human nervous system most likely results in many pathological changes, including its agonistic properties to glutamate and disruption of glutamate receptors. Additionally, this compound potentially causes oxidative stress and impairs cell proliferation [21,22,23].

DA also has high excitatory potential, acting primarily in the central nervous system and myocardium, where it reacts with propanoic acid receptors and kainate subclasses of glutamate receptors. The impact of DA on neuronal cells causes the accumulation of Ca^2+^ and Na^+^, leading to long-lasting depolarization. Moreover, this compound contributes to cellular changes, including mitochondrial and DNA damage, increased ROS concentration and oxidative stress. It has been shown that DA can also impair the function of other organs such as the liver or kidneys. In 1987 four people died after consuming mussels contaminated with DA for review: [24].

In recent decades, efforts of many scientific groups have concentrated on the toxic properties of cyanotoxins with special emphasis on changes in antioxidant systems. Available data in this area are relatively broad and contain information about enzymes involved during oxidative stress or antioxidants scavenging free radicals. However, knowledge of the detailed mechanisms of cyanotoxins’ impact, including disrupted metabolic pathways in the context of antioxidant systems, is still scarce. It was demonstrated that some other toxins such as arsenic species or formaldehyde might interfere with a crucial antioxidant system component, glutathione (GSH), in a potentially direct reaction, which can lead to harmful effects at the cellular level [25,26]. In our recent paper, we examined the influence of CYN or its decomposition products on GSH activity in a direct reaction. Obtained results suggest that neither the toxin nor its decomposition products did not affect GSH in vitro activity against free radicals [27]. To our best knowledge, there is no similar information about other cyanotoxins; therefore, in this paper, we examined the in vitro antioxidant properties of GSH after its direct reaction with toxins produced by cyanobacteria: MC-LR, NOD, ANTX-a, BMAA; and neurotoxin synthesized by diatoms (DA).

## 2. Materials and Methods

### 2.1. Source of Toxins

The source of microcystin-LR was *Microcystis aeruginosa* (Kützing) PCC 7813 from Institut Pasteur (Paris, France), and ANTX-a was obtained from *Dolichospermum flos-aquae* (Lyngb.) de Bréb. (*Anabaena flos-aquae*) strain SAG 30.87 purchased from the University of Göttingen (Göttingen, Germany). NOD, DA and BMAA were purchased from Merck Millipore (Burlington, MA, USA). Cyanotoxins were dissolved in Milli-Q water before use in experiments.

### 2.2. Cyanobacterial Growth Conditions

Axenic cultures of *D. flos-aquae* and *M. aeruginosa* were cultivated separately in BG11 medium [28] in a phytotron at 22 ± 1 °C with 80 ± 5% humidity and 25 µmol m^−2^s^−1^ photosynthetically active radiation (PAR) under a 12 h light/12 h dark photoperiod (lamps AQUAEL 18 W PLANT). All cultures were shaken once a day for 10 min.

### 2.3. MC-LR and ANTX-a Extraction and Purification

After 1 month of cultivation, the cyanobacterial cells were separated from the medium by filtration through GF/C glass microfiber filters (Whatman, UK). In the next step, the isolated cells were frozen at −20 °C and lyophilized. MC-LR and ANTX-a were extracted from the cell biomass according to the method described by [29]. MC-LR and ANTX-a were purified from the cell extract by ultra-high performance liquid chromatography (UHPLC). Briefly, Shimadzu Nexera-I LC-2040C 3D Plus was used. The gradient mobile phase consisted of water/acetonitrile (both acidified with 0.05% trifluoroacetic acid), where the organic phase increased from 2% to 90% over 15 min at a flow rate of 0.75 mL·min^−1^. Samples were separated on a Gemini^®^ NX-C18 Column (110 Å, 3.0 µm, 150 mm × 4.6 mm, Phenomenex, Torrance, CA, USA) maintained at 40 °C. The Autosampler cooler temperature was 4 °C, and the PDA cell temperature was 40 °C. Toxins were identified by comparing the retention time, and UV-spectra determined for commercial standards and quantified by absorbance at 239 and 227 nm for MC-LR and ANTX-a, respectively. A multilevel calibration curve was obtained using commercial standards (from 0.01 to 10.00 µg·mL^−1^).

The presence of toxin in samples was confirmed by using an ultra-performance liquid chromatography tandem-mass spectrometer (UPLC-MS/MS) coupled with a Waters TQD mass spectrometer (electrospray ionization mode ESI-tandem quadrupole) according to the method described in detail by [30,31].

### 2.4. Determination of In Vitro Antioxidant Properties of GSH under the Influence of Cyanotoxins and DA

Antioxidant properties of GSH were measured using a free radical 2.2-Diphenyl-l-picrylhydrazyl (DPPH), according to the spectroscopy method described by [32] with some modifications. The DPPH scavenging activity assay allowed the rapid quantification of antioxidant capacity of single compounds or biological material such as cells or tissues extracts, foods, etc. DPPH (deep blue in colour) abstracts a hydrogen atom in a one-electron reaction to form 2,2-Diphenyl-1-picrylhydrazine (DPPH-H) (pale yellow).

During the reduction of DPPH by compounds that possess antioxidant properties, color change and a decrease in absorption (at λ = 517 nm) are observed. The decrease of absorption expressed as a concentration (%) of remaining DPPH is a result and simple way to evaluate the antioxidant capacity of a sample [31].

In the first step, the antioxidant properties of pure GSH (aqueous solution; 2, 5, and 10 mM) were measured. The reaction mixture contained the following components: 1.5 mL of DPPH ethanolic solution (0.5 mM) and 200 µL of GSH in appropriate concentration. The reduction of DPPH was monitored over 30 min in 3 min intervals at λ = 517 nm.

In the second step, in order to determine the influence of MC-LR, NOD, ANTX-a, BMAA or DA (final concentration: 1, 5 and 10 µg·mL^−1^) on in vitro GSH activity, cyanotoxins were co-incubated with GSH (200 µL; 2, 5 and 10 mM) in darkness at 21 ± 1 °C for 2 h. After 2 h, DPPH (1.5 mL, 0.5 mM) was added to reaction mixtures, and the reduction was measured as detailed above.

Additional measurements with pure cyanotoxins or DA and without GSH were also developed in order to evaluate any potential antiradical properties, which could confuse the interpretation of the results. The reaction mixture was as follows: DPPH (1.5 mL; 0.5 mM) and MC-LR, NOD, ANTX-a, BMAA or DA (final concentration: 1, 5 and 10 µg·mL^−1^). All measurements were performed by using spectrophotometer Jasco V-650 (Jasco Inc., Tokyo, Japan).

### 2.5. Statistical Analysis

All data are expressed as the mean ± standard deviation (SD) of three independent replicates. All results were subjected to ANOVA with *p* < 0.05.

### 2.6. Chemicals

All reagents used in experiments were analytical, HPLC or MS grade and were purchased from Merck Millipore (Burlington, MA, USA).

## 3. Results and Discussion

The concentrations of GSH used in this study reflect the GSH concentrations commonly found in the cytosol of cells [33]. Pure GSH at 2, 5 and 10 mM decreased the concentration of DPPH within 30 min by 45%, 57% and 78%, respectively (Figure 2A).

Results obtained for pure GSH confirmed previously described high antioxidant properties of this compound [34,35]. The biochemical role of GSH in living cells is broad and essential for proper functioning. This tripeptide is a crucial part of antioxidant systems acting during the scavenging of ROS as well as reactive nitrogen species (RNS). Disruption of GSH functioning and/or its synthesis are processes often attributed to cyanotoxins [36,37,38]. To date, however, information about any potential, direct effect of cyanotoxins on GSH is limited. In many cases, consequences of cyanotoxin action resulting in depletion of GSH concentration or increasing the activity of the enzymes involved in its synthesis are presented without a description of the mechanism. The main aim of our study was to concentrate on a single property of GSH—its potential to scavenge free radicals under the influence of certain cyanotoxins and DA. In our opinion, broadening the knowledge in this area could provide valuable data about the impact of cyanotoxins and DA in the context of direct GSH activity.

The results presented in this paper suggest that MC-LR, NOD, ANTX-a, BMAA and DA do not affect the ability of GSH to scavenge free radicals directly. None of the compounds disrupted the potential for GSH to scavenge DPPH in vitro (Figure 2B–F—data for the highest concentration of GSH used in the experiments).

Several studies indicated that the impact of MC-LR on the antioxidant defence system present in animal cells is strong and occurs both in the natural aquatic environment and under laboratory conditions [39,40].

Sicińska, P. 2006 [41] showed that MC-LR increases the in vitro ROS concentration and changes the activity of antioxidant system enzymes: catalase (CAT), glutathione reductase (GR) and superoxide dismutase (SOD) in human erythrocytes. They postulated that the presence of observed damages in the cell membrane and in studied enzymes could be the indirect result of oxidative stress caused by MC-LR. [40] investigated the influence of pure MC-LR and MC containing crude extract on the human colon carcinoma cell line Caco-2. The activity of CAT, GR, glutathione peroxidase (GPX), glutathione-S-transferase (GST) and SOD were analyzed. Moreover, studies on lipid peroxidation, ROS and GSH were also performed. The results obtained suggest that the extract containing MC-LR was the most toxic in these tests, followed by pure MC-LR. The CAT, GR, ROS and SOD were treated as oxidative stress biomarkers, and its activity under MC-LR exposure was more highly disrupted, but, potentially, compounds present in crude extract could increase the intensity of the influence of the toxin [40]. Martins, N.D 2017 [42] showed that four species of fish collected from a lake covered by a bloom of MC-LR producers had biochemical and physiological changes similar to those observed under laboratory conditions. More evidence of the harmful impact of MC-LR in the context of antioxidant systems in the natural environment is available [39,43,44].

The research described above, as well as other research, suggest that the impact of MC-LR on animal cells under both laboratory conditions and in nature are relatively similar and changes in concentration of GSH is often the observed effect. [45] described the potential process of MC-LR detoxification and demonstrated that it could be conjugated to GSH by GST in rat cells. Earlier studies postulated that MC-LR could conjugate directly to GSH, but this process most likely depends on pH conditions and needs special conditions (the presence of potassium carbonate) [46]. [47] also showed that GSH metabolism is crucial during MC-LR detoxification in mouse liver cells. They noted increased activity of enzymes involved in antioxidant response (GPX and GST) and the increased concentration of GSH. [48] studied the toxicity of MC-LR both under the influence of free GSH and after its conjugation to GSH (MC-LR-GSH) on human hepatoma HepG2 cells. Results showed that the bioactivity of MC-LR acting with free GSH was lower, whereas the toxicity of the MC-LR-GSH conjugate was significantly decreased compared to pure MC-LR. The MC-LR-GSH conjugate was chemically synthesized and could also be formed at a cellular level by the action of glutathione transferases. However, results focussing on the toxicity of MC-LR in the presence of free GSH showed that the formation of the MC-LR-GSH conjugate is relatively complicated and not spontaneous. Numerous data available for GSH metabolism under the influence of MC-LR do not answer the important questions regarding whether the antioxidants properties of this compound are changed. Without a doubt, the preservation of in vitro GSH antioxidant properties under the influence of MC-LR could be treated as a valuable feature of this tripeptide that also potentially occurs at a cellular level beside other GSH-metabolism pathways. Parts of our results focus on the interaction between MC-LR and GSH and provide additional information to the study referred to above, leading to conclusions that this toxin probably does not disrupt the in vitro activity of GSH.

Available data about NOD influence on GSH metabolism are generally consistent with those obtained for MC-LR. Based on the observation of similar chemical structures, NOD influence on GSH metabolism in living organisms could be similar to that of MC-LR. [49] demonstrated that NOD is also conjugated to GSH by GSTs in brine shrimp *Artemia salina* cells. The authors indicated that it could be one of the first stages of NOD detoxification. The preservation of the antiradical potential of GSH under the influence of NOD undoubtedly facilitates the neutralization of poisoning. It seems to be especially important in the context of data referring to increasing ROS concentration in animal cells under even low NOD concentration [50].

Information about ANTX-a interaction with GSH in animal cells is limited. However, there is some information about the impact of this toxin in plant cells, but it is primarily about changes in the enzymes involved in GSH metabolism [51,52,53,54]. Our results suggest no disruption of GSH antioxidant properties by ANTX-a. However, even if there was an interaction between the two, then the fast action of ATNX-a in animal cells as an acetylcholine agonist would overshadow any effect of it on GSH.

Results obtained for another neurotoxin, BMAA, were similar. It has been documented that BMAA is an agonist of glutamate and causes the overstimulation of glutamate receptors. Moreover, its action in animal cells, especially in motor neurons, causes increased ROS concentration leading to oxidative stress and disruption of Ca^2+^ ion concentration [21,55]. Based on the effect of BMAA on animal cells, it could be assumed that it most likely stimulates the antioxidant activity of GSH. Results presented in this paper showed that in a direct reaction, BMAA did not disrupt GSH activity. However, some studies proposed that the bioactivity of this toxin involved the depletion of the GSH concentration through the inhibition of the cystine/glutamate antiporter system Xc^−^ [56].

Similarly, we did not observe disruption of GSH antioxidant properties under the influence of DA. [57] linked the toxicity of DA with GSH metabolism in mice cerebellar granule neurons (CGNs). In cells without the modifier subunit of glutamate-cysteine ligase (an enzyme involved in GSH synthesis), apoptosis induced by DA developed faster in comparison to the controls. Conversely, DA caused the oxidative stress and efflux of GSH [57]. The role of GSH in the toxicity of DA in the context of oxidative stress was later confirmed. Mice CGNs possessing a system of GSH synthesis and physiological concentration of this antioxidant were less sensitive to apoptosis and necrosis induced by oxidative stress caused by DA [36]. Our results showed that the properties of DA do not include directly affecting the scavenging of radicals by GSH in vitro. Potentially, GSH could act similarly under the influence of DA at the cellular level to the moment of the development of other toxic effects and its efflux from cells.

Knowledge about the impact of cyanotoxins and DA on living organisms or cells is relatively broad. Information about its influence on many metabolic pathways, changes in concentration of certain enzymes or modulators and consequences of its action are available in many works. However, detailed data on the impact of cyanotoxins at the cellular level are often not available. Experiments carried out under in vitro conditions that concentrate on a single reaction or metabolism of single compounds under the influence of cyanotoxins are sources of new information that could be useful during in vivo studies or the interpretation of environmental data. In this work, we adapted, for cyanotoxins and DA research, a very sensitive and unequivocal assay that allowed us to investigate their impact on in vitro GSH antioxidant properties. Undoubtedly, GSH and its metabolism are an important part of the system preventing toxicity of cyanotoxins, and its disruption largely contributes to harmful effects caused by these compounds. The results described in this work complement those obtained by other authors and shows that the analyzed toxins do not affect GSH activity directly. In our opinion, further studies should focus on in vivo studies on the GSH antioxidant system under the influence of cyanotoxins and DA at the cellular level.

## 4. Conclusions

The key outcomes of this study are that: (i) cyanobacterial hepatotoxins, e.g., MC-LR and NOD, do not influence the in vitro antioxidant potential of GSH; (ii) similarly, neurotoxins produced by cyanobacteria, e.g.,ANTX-a, BMAA and by diatoms such as DA do not influence in vitro GSH antioxidant properties; (iii) obtained results provide new and valuable information about GSH metabolism under the influence of cyanotoxins and DA and are the first description of in vitro GSH antioxidant properties after direct reaction with MC-LR, NOD, ANTX-a, BMAA and DA; (iv) as postulated by other authors, the impact of cyanotoxins and DA on GSH activity can take place indirectly.

## Figures and Tables

**Figure 1 life-12-00227-f001:**
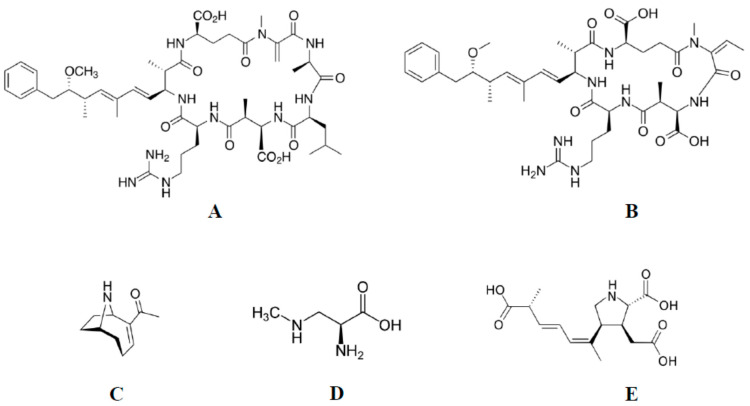
Chemical structures of: microcystin-LR (**A**), nodularin (**B**), anatoxin-a (**C**), β-N-methylamino-L-alanine (**D**) and domoic acid (**E**).

**Figure 2 life-12-00227-f002:**
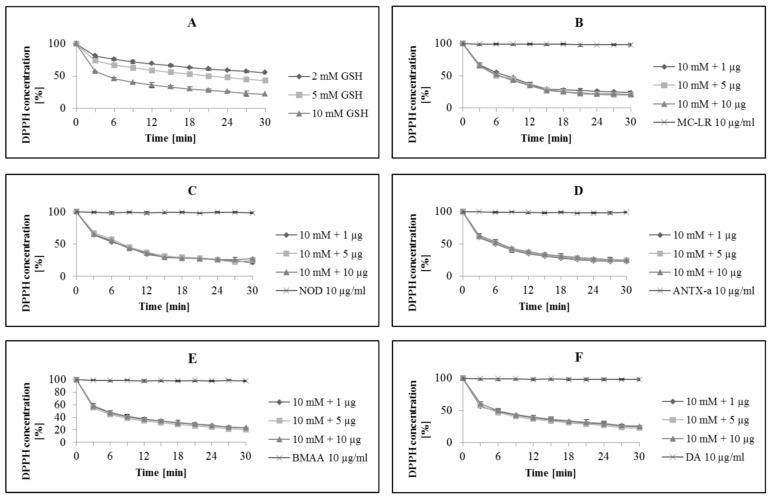
Kinetic changes in DPPH concentration under the influence of: pure glutathione (GSH) (**A**); GSH (10 mM) after co-incubation with: microcystin-LR (**B**), nodularin (**C**), anatoxin-a (**D**), β-N-methylamino-L-alanine (**E**) and domoic acid (**F**). The final concentration of toxins (µg·mL^−1^) was shown as 1 µg, 5 µg and 10 µg. MC-LR 10 µg/mL, NOD 10 µg/mL, ANTX-a 10 µg/mL, BMAA 10 µg/mL, DA 10 µg/mL—negative control for single toxins. Data are means ± SD of three independent replicates.

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
