# Peer review of "Effect of Microcystin-LR, Nodularin, Anatoxin-a, β-N-Methylamino-L-Alanine and Domoic Acid on Antioxidant Properties of Glutathione"

_life, 2022, doi:10.3390/life12020227_

Round 1

Reviewer 1 Report

The authors addressed the majority of the comments, and executed relevant changes to the manuscript that did improve it significantly. However, in my opinion the introduction should instead of describing cyanotoxins in a very broad way, focus on the oxidative stress inducing properties of the tested cyanotoxins and DA in combination with the cellular oxidative stress response, explaining the role of GSH, the effects of the tested compounds have been reported to induce with regard to GSH on the cellular level, and why this study, addressing the direct interaction of the selected cyanotoxins and DA, is relevant in this context. The discussion was improved and methods are sufficiently described.

Grammar was improved, however there are still some grammatically incorrect or confusing sentences, missing articles and typos. I suggest an additional minor spellcheck of the whole manuscript.

Abstract: 

  • The first sentence contains substantially basic knowledge that in my opinion does not belong in an abstract in my opinion.
  • Line 21: (DA) should be moved behind the word “acid”.

Introduction:

  • There is a paragraph in the introduction dedicated to the description of cylindrospermopsin, which is not studied in this manuscript. This section should be removed, as well as the section about dermatotoxins in general.
  • Line 50-51: “organ death” should be changed to “organ failure”.
  • The sentence in line 104-106 should be grammatically corrected.
  • Line 107: in vitro should be italic

Results and discussion:

  • Figure 2: The legends should be modified to be more explanatory or the legend should be clearly explained in the figure caption.
  • The discussion is still lacking relevant (newer) references.
  • In my opinion the discussion should include only studies on pure toxins and not cyanobacterial extracts containing cyanotoxins as there are a myriad of substances present that can influence the induction of oxidative stress, as also pointed out by the authors. However, such information should not be added to the discussion as it is irrelevant for the context of this manuscript, dealing with direct interactions of pure cyanotoxins with GSH.

I would like to briefly comment on the suggestion for the need for further in vivo studies. In my opinion in vivo studies should not be conducted unless there is very valid reason and substantial benefit can be expected from that. There is already a great amount of knowledge about the mechanisms of action of the studied cyanotoxins and about their influence on the oxidative response pathways. Therefore suggesting to conduct in vivo studies should not be done lightly and should involve great consideration, also given that no direct influence on the GSH antioxidative properties could be detected in vitro.

Author Response

Dear Editor,

            We are pleased to re-submit our manuscript life-1504549 entitled “Effect of microcystin-LR, nodularin, anatoxin-a, β-N-methylamino-L-alanine and domoic acid on antioxidant properties of glutathione” after revision. We would like to thank you and the Reviewer for your and his time and all comments. All changes are presented below.

Response to Reviewer comments

Reviewer: The authors addressed the majority of the comments, and executed relevant changes to the manuscript that did improve it significantly. However, in my opinion the introduction should instead of describing cyanotoxins in a very broad way, focus on the oxidative stress inducing properties of the tested cyanotoxins and DA in combination with the cellular oxidative stress response, explaining the role of GSH, the effects of the tested compounds have been reported to induce with regard to GSH on the cellular level, and why this study, addressing the direct interaction of the selected cyanotoxins and DA, is relevant in this context. The discussion was improved and methods are sufficiently described.

Grammar was improved, however there are still some grammatically incorrect or confusing sentences, missing articles and typos. I suggest an additional minor spellcheck of the whole manuscript.

Response: We are very grateful for the detailed comments and suggested improvements. Our idea for introduction was to presented cyanotoxins and DA as harmful compounds impairing many metabolic pathways in living cells. We tried to link this information with information of efforts of scientific groups for broaden the knowledge of cyanotoxins impact on antioxidant system. We performed study concentrate on relatively detailed issues and trying show it against more related in the discussion sections. However, we removed parts of cylindrospermopsin and dermatotoxins – cyanotoxins not studied in this work. The manuscript text was grammar spellchecked by professional native speaker, list of references was updated. Below we presented other responses for your comments.

Reviewer: Abstract:

Reviewer: The first sentence contains substantially basic knowledge that in my opinion does not belong in an abstract in my opinion.

Response: This sentence was modified.

Reviewer: Line 21: (DA) should be moved behind the word “acid”.

Response: This sentence was corrected.

Reviewer: Introduction:

Reviewer: There is a paragraph in the introduction dedicated to the description of cylindrospermopsin, which is not studied in this manuscript. This section should be removed, as well as the section about dermatotoxins in general.

Response: These parts were removed according to your suggestion.

Reviewer: Line 50-51: “organ death” should be changed to “organ failure”.

Response: This term was modified.

Reviewer: The sentence in line 104-106 should be grammatically corrected.

Response: This sentence was modified.

Reviewer: Line 107: in vitro should be italic

Response: This term was modified.

Reviewer: Results and discussion:

Reviewer: Figure 2: The legends should be modified to be more explanatory or the legend should be clearly explained in the figure caption.

Response: The figure caption was modified according to your comment.

Reviewer: The discussion is still lacking relevant (newer) references.

Response: There is no other works showed influence of cyanotoxins on GSH activity in direct reaction. Therefore, we try to choose the works contains information that are the most related to our studies. We introduce new parts in the discussion section and use new references. We try to choose works that for the first time presented related to our issues. Please note that 1/3 of all references are works published in the last six years.

Reviewer: In my opinion the discussion should include only studies on pure toxins and not cyanobacterial extracts containing cyanotoxins as there are a myriad of substances present that can influence the induction of oxidative stress, as also pointed out by the authors. However, such information should not be added to the discussion as it is irrelevant for the context of this manuscript, dealing with direct interactions of pure cyanotoxins with GSH.

Response: Please note that part of the discussion including information about cyanobacterial crude extract is linked with data about pure MC-LR. In our opinion fact that compounds presents in cells extract could increase the effect of single toxin is important and potentially useful for readers. We would prefer keep this part in the text of manuscript.

I would like to briefly comment on the suggestion for the need for further in vivo studies. In my opinion in vivo studies should not be conducted unless there is very valid reason and substantial benefit can be expected from that. There is already a great amount of knowledge about the mechanisms of action of the studied cyanotoxins and about their influence on the oxidative response pathways. Therefore suggesting to conduct in vivo studies should not be done lightly and should involve great consideration, also given that no direct influence on the GSH antioxidative properties could be detected in vitro.

Response: We totally agree with you. However, studies concentrate of influence of cyanotoxins under in vivo conditions potentially will be undertaking in the future. Perhaps our work will be motivation for performed additionally tests allow to broad the knowledge about impact of cyanotoxins on GSH in the direct reaction. We have the same opinion for not abusing suggestions about in vivo tests in publications, but also in the reviewers or editors comments.

Thank you very much for your time. We hope that all explanations are satisfactory and our paper will receive your permission for publication in Life.

Yours faithfully,

Reviewer 2 Report

The manuscript comes with an evaluation of direct effects of toxins produced by cyanobacteria: MC-LR, NOD, ANTX-a, BMAA; and neurotoxin synthesized by diatoms- DA on in vitro antioxidant properties of glutathione (GSH). The obtained results addressed that all the studied toxins had no effect on GSH antioxidant potential. The authors concluded about the importance of in vivo studies for the evaluation of cyanotoxins and DA effects on GSH properties.

The manuscript has to be improved in order to be suitable for publication.

To improve the manuscript, authors should consider the following recommendations:

- Authors must justify the GSH concentration used for the conducted test.

- Authors provided in the results section only data about 10 mM of GSH added to toxins (cyanotoxins or DA).

They must indicate why the results of the other GSH concentrations were not cited; since they used 2, 5 and 10 mM (Line 162) to test the effects of toxins on GSH antioxidant potential.

- To give more coherent manuscript, authors could include to the proposed study an in vivo test to support the provided results and enrich the version of the manuscript with a comparison of in vitro and in vivo effects of those toxins on GSH properties.

- English spell check is required

- Reference citation must meet with the journal instructions to the authors:

 * References in the text must be cited using numbers.

* The list of the references at the end of the manuscript must respect the journal guidelines.

Author Response

Dear Editor,

            We are pleased to re-submit our manuscript life-1504549 entitled “Effect of microcystin-LR, nodularin, anatoxin-a, β-N-methylamino-L-alanine and domoic acid on antioxidant properties of glutathione” after revision. We would like to thank you and the Reviewer for your and his time and all comments. All changes are presented below.

Response to Reviewer comments

Reviewer: The manuscript comes with an evaluation of direct effects of toxins produced by cyanobacteria: MC-LR, NOD, ANTX-a, BMAA; and neurotoxin synthesized by diatoms- DA on in vitro antioxidant properties of glutathione (GSH). The obtained results addressed that all the studied toxins had no effect on GSH antioxidant potential. The authors concluded about the importance of in vivo studies for the evaluation of cyanotoxins and DA effects on GSH properties.

The manuscript has to be improved in order to be suitable for publication.

To improve the manuscript, authors should consider the following recommendations:

Response: We are very grateful for the detailed comments and suggested improvements. All responses for your comments are presented below.

Reviewer: Authors must justify the GSH concentration used for the conducted test.

Response: The concentrations of GSH used in this study reflect the GSH concentrations commonly found in the cytosol of cells (for review: Forman et al. 2009; Meister 1988). This information was added in the results and discussion section.

Reviewer: Authors provided in the results section only data about 10 mM of GSH added to toxins (cyanotoxins or DA). They must indicate why the results of the other GSH concentrations were not cited; since they used 2, 5 and 10 mM (Line 162) to test the effects of toxins on GSH antioxidant potential.

Response: In the text of the manuscript is information that: “None of the compounds disrupted in vitro GSH potential to scavenge DPPH (Fig. 2 B, C, D, E and F – data for the highest concentration of GSH used in the experiments).” We decided to show only data for the highest tested GSH concentration – the graphs for other applied GSH concentration correspond with it. Please note that studies presented in this work correspond with earlier studies [Adamski, Michal, & Kaminski, A. (2021). Impact of cylindrospermopsin and its decomposition products on antioxidant properties of glutathione. Algal Research, 56] and we decided to keep similar graphic presntation of the results. However, we expanded the caption of the Figure 2.

Reviewer: To give more coherent manuscript, authors could include to the proposed study an in vivo test to support the provided results and enrich the version of the manuscript with a comparison of in vitro and in vivo effects of those toxins on GSH properties.

Response: Definitely, in vivo tests would support presented results and it will be our aim for future projects. We suggest important role of the in vivo tests in the context of our studies in the results and discussion section.

Reviewer: English spell check is required

Response: Text of the manuscript was checked by professional native speaker according to your suggestion.

Reviewer: Reference citation must meet with the journal instructions to the authors:

* References in the text must be cited using numbers.

* The list of the references at the end of the manuscript must respect the journal guidelines.

Response: Please note, that template of the manuscript was created by the journal office and there was not recommendations for the scheme of the text. Of course if we receive some recommendations in the future we introduce all of them.

Thank you very much for your time. We hope that all explanations are satisfactory and our paper will receive your permission for publication in Life.

Yours faithfully,

Round 2

Reviewer 2 Report

The authors have considered the comments of the reviewers and have included the remarks and modifications. The main concerns of the manuscript have been solved. In my opinion the provided version is now suitable for publication